

# Using Data Assimilation to Improve Data-Driven Models.

Michael Goodliff[1], Takemasa Miyoshi[1]

[1]RIKEN Center for Computational Science, Kobe, Japan

*Correspondence to*: Michael Goodliff (michael.goodliff@riken.jp)

**Abstract.** Data-driven models (DDMs) are developed by analysing extensive datasets to detect patterns and make predictions, without relying on predefined rules or instructions from humans. In fields like numerical weather prediction (NWP), DDMs are gaining popularity as potential replacements for traditional numerical models, thanks to their grounding in a multi-decadal, high-quality data assimilation (DA) analysis product. Recent studies, such as Lang et al. (2024), have demonstrated that training DDMs using the ERA5 (the fifth generation European Centre for Medium-range Weather

Forecast atmospheric reanalysis) can outperform traditional numerical models. DA integrates observations from various sources with numerical models to enhance the accuracy of model state estimates and predictions or simulations of a system's behaviour. Due to the benefits of DDMs and DA, integration of these methods has been gaining traction in a wide range of fields.

This paper focuses on the application of DA methodologies in enhancing the precision and efficiency of DDM generation. The aim is to demonstrate the pivotal role that DA can play in refining and optimising the process of DDM generation by incorporating various observation data directly, augmenting the accuracy and reliability of predictive models despite the presence of observational uncertainties. This study shows how DDMs can improve on imperfect model forecasting, and in conjunction with DA, can cyclically generate more accurate training data, further enhancing the precision of DDMs.

## 1 Introduction


Data-driven models (DDMs) have become increasingly popular in many fields over the past decade. These models rely on data to make predictions, decisions, or generate insights. Unlike traditional models that are often based on theoretical frameworks or assumptions, DDMs are built and refined by using machine learning (ML) techniques on large datasets to detect patterns and make predictions. DDMs can outperform traditional models in the field of numerical weather prediction

(NWP) as shown by Bouallègue (2024) and Olivetti (2024) and are generally computationally cheaper to run. Bouallègue (2024) compared ML generated forecasts with NWP systems in an operational-like setting and found that ML models show overall good performance against the operational Integrated Forecasting System (IFS) but with some unphysical representations (Bonavita, 2024). In this study we attempt to improve on this DDM generation with the use of cycling data assimilation (DA) and ML. DA is a mathematical framework for combining observational data with numerical models to

improve estimates of a system's state, often used in NWP and environmental sciences. It leverages techniques such as



Bayesian inference, variational methods, and ensemble approaches to optimally merge uncertain observations with prior knowledge from numerical models.

To integrate DA and ML for generating more accurate DDMs, we first construct an imperfect model analysis trajectory - an imperfect model prediction that is corrected/updated using DA with noisy and partial observations. A DDM is then trained on this analysis trajectory, aiming to approximate the system dynamics more accurately. This trained DDM is used for DA with the same observations to construct a new analysis trajectory, which would be more accurate if the DDM is more accurate than the imperfect model. Then, the DDM trained upon the new analysis trajectory could be more accurate. This iterative process continues, we would expect progressively refining the accuracy of the DDMs.

The combination of DA and DDMs has gained recognition due to how this integration of methods can improve time series forecasting. This study aims to combine DA and DDMs for improving the accuracy of the DDMs themselves.

An example of combing DA and ML comes from Brajard (2019). They created a method to generate DDMs using an iterative algorithm from partial and noisy observations on the single-scale Lorenz 96 model. This method uses DA to iteratively interpolate and de-noise the observations, the ML will then learn the dynamics of the DA analysis. However, they did not discuss how an imperfect model could be improved.

Another example of using DA and ML together can be seen in Amemiya (2023). On the two-scale Lorenz-96 model, ML is used to correct systematic model biases through DA. They applied neural networks to model bias correction through DA in idealised experiments with noisy and sparse observations. Among the different neural network architectures tested, recurrent neural networks (RNNs) performed the best in correcting systematic model bias and enhancing extended forecast accuracy. The findings indicate that incorporating past time series of forecast variables helps improve model bias correction, especially when only limited observational data is available for assimilation. As per the results given in Amemiya (2023), our experiments will use the Long-Short Term Memory (LSTM) as our ML methodology. An LSTM is a specialised RNN that learns and retains long-term dependencies in time series data effectively. This is explained more thoroughly in the Materials and Methods section.

The method presented in this paper corrects imperfect model dynamics by incorporating noisy, partial observations of the perfect system, capturing unseen dynamics that can influence model forecasting.

## 2 Methodology

### 2.1 Lorenz 63 Systems

In this paper, we use two version of the Lorenz 63 model. The first is the traditional, dynamical model which is designed to





produce chaotic behaviour to mimic atmospheric convection (L63, Lorenz (1963)). The second model is the "coupled

chaotic" Lorenz model which come from coupling the Lorenz model with a linear oscillator, this model is used to mimic

extratropics coupled with the tropics (L63-cc, Palmer (1993)).

These model equations are given by:

$$\frac{dx_0}{dt} = -\sigma(x_0 - x_1),$$

$$\frac{dx_1}{dt} = \rho x_0 - x_1 - x_2 x_0,$$

$$\frac{dx_2}{dt} = x_0 x_1 - \beta x_2,$$

and

$$\frac{dx_0}{dt} = -\sigma(x_0 - x_1) + x_4,$$

$$\frac{dx_1}{dt} = \rho x_0 - x_1 - x_2 x_0 + x_3,$$

$$\frac{dx_2}{dt} = x_0 x_1 - \beta x_2,$$

$$\frac{dx_3}{dt} = -\omega x_4 - k(x_3 - x_3^*) - x_1,$$

$$\frac{dx_4}{dt} = \omega(x_3 - x_3^*) - k x_4 - x_0,$$

where $x_{\{0..5\}} = x_{\{0..5\}}(t)$ are the state variables (where t is time) and $\sigma = 10$, $\rho = 28$, $\beta = 8/3$,

$\omega = 1.5$, and k=0.1 are parameters. In Palmer (1993), $x_3^*$ represents an El Nino SST anomaly and is set to zero here.

**2.2 Three-Dimensional Variational Data Assimilation (3DVar)**

In the realm of numerical weather prediction and DA, Three-Dimensional Variational DA, or 3DVar (Lorenc (1986), Kalnay

(2002)), serves as a foundational method for combining observational data with prior estimates of a system's state. Its

principal aim is to produce the most likely estimate of the system state by finding a balance between these two sources of

information, considering their respective uncertainties.

At the heart of 3DVar lies a cost function $J(x)$, which represents the balance between the "background" (the prior state) and

the observations. The mathematical formulation of this cost function is as follows:

$$J(x) = (x - x_{\mathrm{b}})^{\mathrm{T}} \mathbf{B}^{-1}(x - x_b) + (y - Hx)^{\mathrm{T}} \mathbf{R}^{-1}(y - Hx)$$

where:





- $x$ is the state vector, representing the unknown system state we aim to estimate.
- $x_b$ is the background state, a prior estimate often derived from a model forecast.
- $y$ is the observation vector, representing measurements of the system. In this experiment, these observations are generated from a Gaussian distribution $N(0, \sigma^2)$ around our perfect model trajectory. Here, we use $\sigma^2 = 1$.
- **B** is the background error covariance matrix, quantifying the uncertainty in the prior estimate $x_b$. In this experiment, B is a climatological matrix generated from taking sample covariances from a long model trajectory. We use the imperfect model to construct matrices for both itself and the LSTM-$n$, while the perfect model generates the matrix for its own use.
- **R** is the observation error covariance matrix, representing the uncertainty in the observations $y$. This covariance matrix in this experiment is the identity which corresponds to $\sigma^2 = 1$.
- $H$ is the observation operator, which maps the model's state $x$ to observation space.

This cost function comprises two terms:

1.  The Background Term:

$$J_b(x) = (x - x_b)^T B^{-1}(x - x_b)$$

This part of the cost function penalises deviations of the estimated state $x$ from the background state $x_b$, weighted by the background uncertainties.

2.  The Observation Term:

$$J_o(x) = (y - Hx)^T R^{-1}(y - Hx)$$

This part of the cost function penalises mismatches between the observations $y$ and their model equivalents $Hx$, weighted by the observation uncertainties.

The optimal state estimate, known as the analysis, is found by minimising $J(x)$. This requires setting the gradient of the cost function to zero:

$$\nabla J(x) = B^{-1}(x - x_b) - H^T R^{-1}(y - Hx) = 0.$$

The solution is found using minimisation methods, such as the conjugate gradient method, which are used iteratively to find the minimum of the cost function.

**2.3 Long-Short Term Memory**

First introduced in 1997 (Hochreiter, 1997), Long-Short Term Memory (LSTMs) are a type of recurrent neural network, renowned for its ability to effectively capture and utilise long-range dependencies in sequential data. At its core, an LSTM consists of memory cells that maintain a hidden state representing the information accumulated from prior inputs. These memory cells are equipped with gating units, including the input gate, forget gate, and output gate.

- The input gate regulates the flow of new information into the memory cell, deciding which parts of the input to incorporate into the current state.
- The forget gate controls the retention or removal of information from the memory cell, determining which previous information to discard.
- The output gate governs the extraction of relevant information from the memory cell to produce the output at the current time step.




These gates are equipped with sigmoid and tanh activation functions, enabling them to adaptively modulate the flow of information based on the input data and the current state of the memory cell. Additionally, the LSTM architecture includes self-connections, allowing the memory cells to retain information over multiple time steps and capture long-term
dependencies in the data.

By dynamically adjusting the gating mechanisms based on the input sequence, an LSTM can effectively learn to recognise patterns, remember important information over extended periods, and make accurate predictions, making it well-suited for a wide range of sequential data processing tasks.


**2.4 Looping Algorithm**

In figure 1, we outline our proposed algorithm to improve data-drive model generation using DA. In this algorithm, we start with an imperfect model. This model is then used with DA on perfect model observations. Using the analysis trajectory, we can generate an LSTM (LSTM-0) to be a better estimate of the system than the imperfect model. If we then repeat this step,
by using the new LSTM instead of the numerical model, this produces a new LSTM (LSTM-1) which is, again, a more accurate representation of the perfect system. This algorithm can be looped $n$ times (here we use $n=30$) to produce a ML model that is closer to the perfect system.

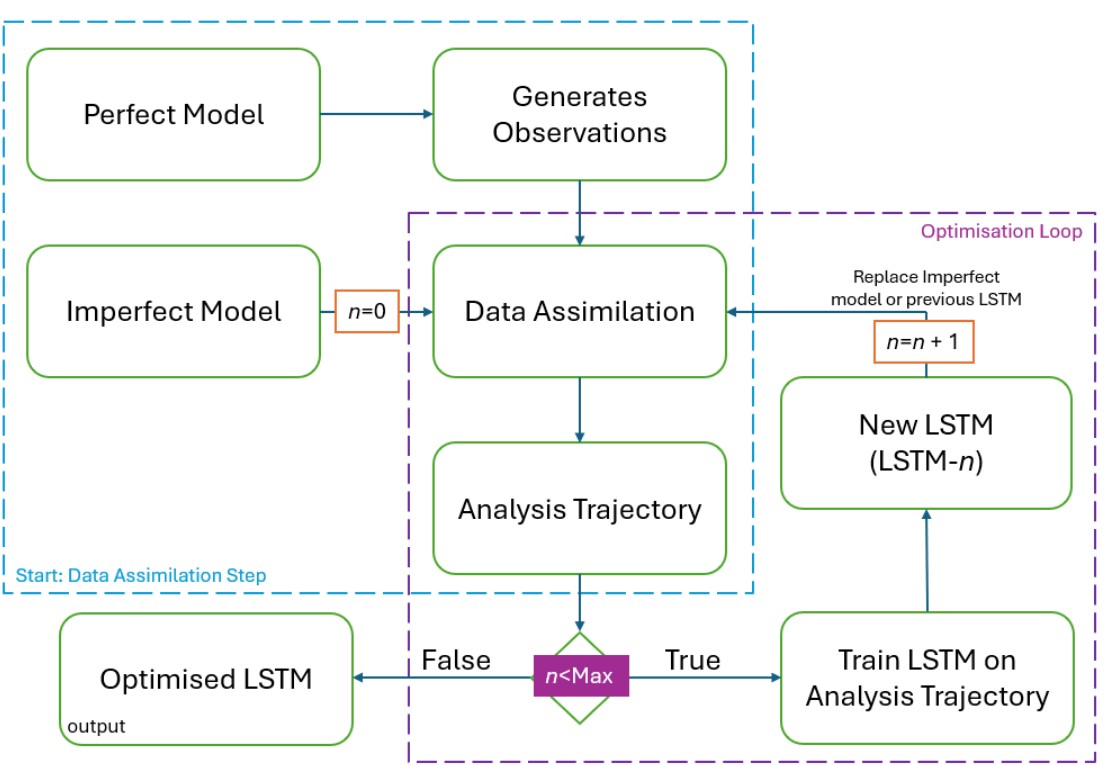




**Figure 1.** This figure shows the cycling algorithm implemented in this paper. The blue dashed box is the traditional DA cycle, the purple dashed box is our additional loop until the max iteration number is reached, giving an optimised LSTM. When *n*=0 indicates the iteration number, LSTM-0 would be the initial LSTM trained on the imperfect model analysis trajectory after its correction using DA.


**Results**

Using the proposed algorithm (detailed in Methodology and Figure 1), we begin with an imperfect model—specifically, the classical Lorenz 63 model with three variables $(x_0, x_1, x_2)$ simulated over a 100,000dt timeframe (dt=0.05). In contrast, the perfect model in our experiments is the coupled chaotic Lorenz 63 system, featuring five variables $(x_0, x_1, x_2, x_3, x_4)$.

Observations of three variables from the perfect model $(x_0, x_1, x_2)$, sampled every 1dt, are combined with the imperfect model using DA to produce an analysis trajectory.

This analysis trajectory is then used to train an LSTM-based DDM configured with 2 layers, 100 neurons, and 500 epochs,
using both training and testing datasets of 100,000dt. The resulting DDM is integrated back into the algorithm and updated

iteratively with DA, incorporating observations from the perfect model. Through this cyclic process, a new DDM is created
at each iteration. The hypothesis is that this iterative approach produces a progressively better estimation of the perfect
model for the observed variables $(x_0, x_1, x_2)$.

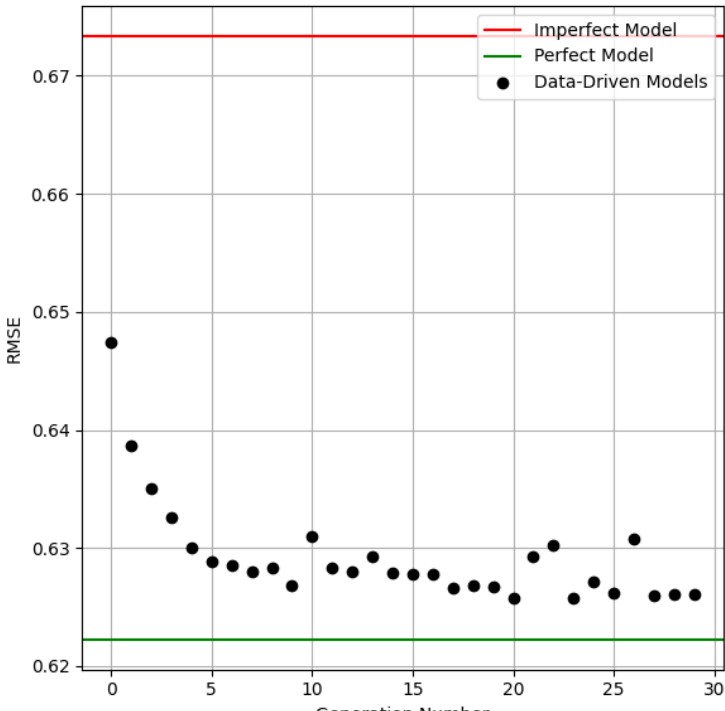


**Figure 2.** This figure shows the analysis RMSE of our cycling algorithm on the Lorenz 63 model. As we cycle through the
algorithm, we see the analysis/forecasts getting closer to the perfect system.

Figure 2 illustrates the analysis RMSEs of our DDMs (LSTMs) compared to the imperfect and perfect models. As expected,

the imperfect model exhibits the highest RMSE, indicating the lowest accuracy, while the perfect model achieves the
greatest accuracy. The x-axis represents successive iterations of the cycling algorithm. The LSTM-0 is trained on the
analysis trajectory of the imperfect model with DA and demonstrates a noticeable improvement over the imperfect model.
The next generation LSTM-1 is trained on LSTM-0 and achieves further reduction in RMSE. After 30 iterations, the RMSE




stabilises, signalling a plateau in performance improvement. This cyclic process highlights the potential of the algorithm to
enhance the accuracy of DDMs.

Figures 3 and 4 explore model forecast error (MFE) for forecasts ranging from 1dt to 4dt. Figure 3 presents the MFE
reduction (%), while Figure 4 visualises its spatial distribution of the MFE. The MFE is calculated by comparing the
imperfect model or LSTM-$n$ against the perfect model. These models all compare forecast trajectories given the same initial
conditions, which are taken from the perfect model long forecast trajectory.

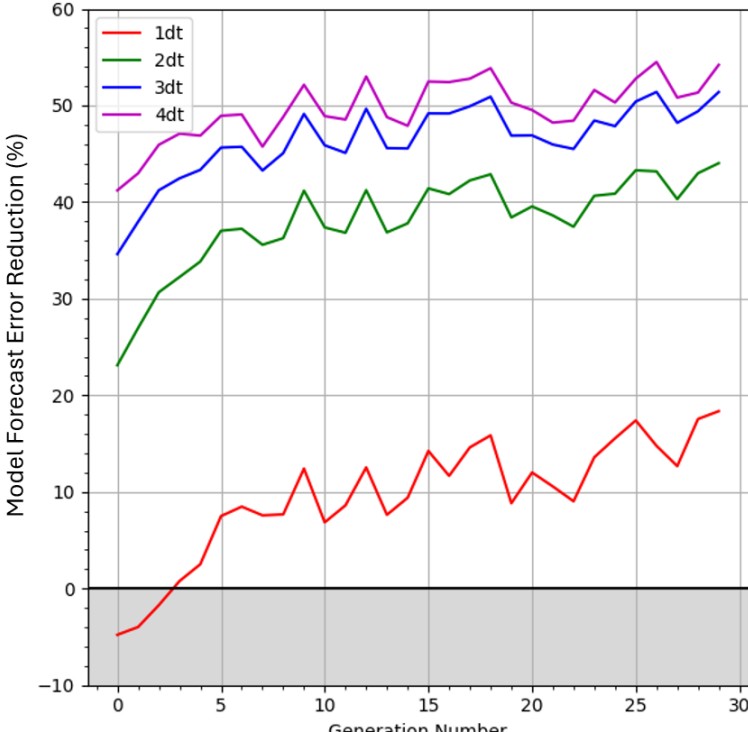

**Figure 3.** This figure displays the model forecast error (MFE) reduction (as a percentage) for forecasts at 1$dt$ (red), 2$dt$
(green), 3$dt$ (blue), and 4$dt$ (purple). The black line represents zero MFE reduction, while the shaded error represents an
increase in MFE.

In Figure 3, the MFE reduction % for each generation (x-axis) of the cycling algorithm is shown. For 1$dt$ forecasts (shown in
red), the initial LSTM iteration is outperformed by the imperfect model. It takes four iterations for the MFE to drop below
the level of the imperfect model. Beyond this point, the algorithm produces a consistent downward trend in MFE, showing
continuous improvement up to the 30-iteration limit. When the forecast length is increased to 2-4$dt$ (green, blue, and purple
lines respectively), LSTM-0 already outperforms the imperfect model across all cases. Subsequent iterations further refine



the forecasts, although the rate of improvement slows as the forecast length increases. Despite this, the overall trend across all forecast lengths demonstrates steady improvement in MFE with each iteration of the algorithm.

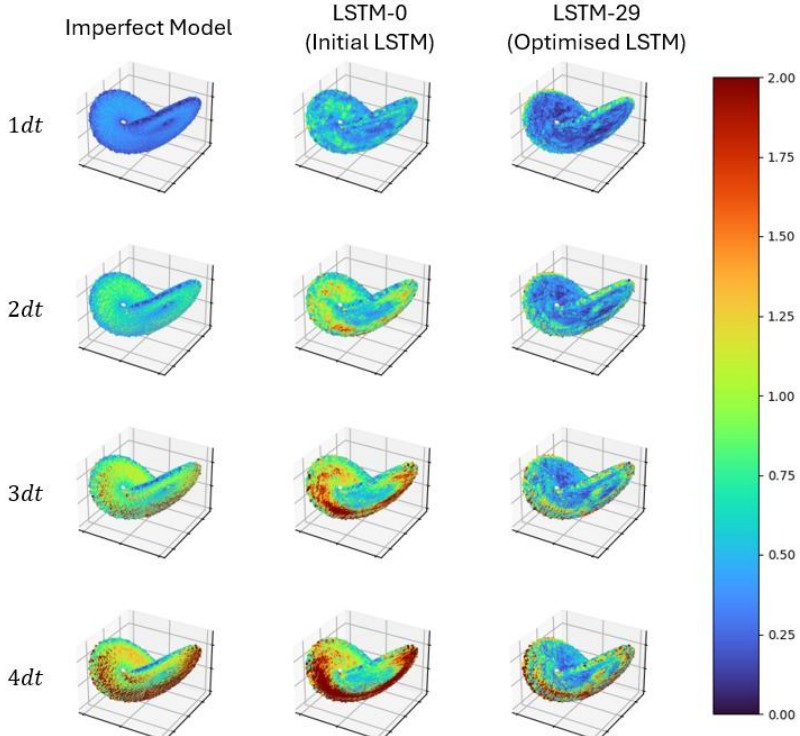

**Figure 4.** This plot illustrates the spatial distribution of forecast errors for 1–4*dt* forecasts. The colours represent the model forecast error for three cases: (1) the imperfect model, (2) the initial LSTM (LSTM-0), and (3) the Optimised LSTM (LSTM-29).

Figure 4 shows the spatial distribution of the MFE for the imperfect model, LSTM-0 (the initial LSTM), and LSTM-29 (the optimised LSTM). In the 1*dt* case, the imperfect model performs notably well compared to the perfect model, particularly along the boundaries of the attractor. Consistent with previous findings, LSTM-0 struggles in this scenario, especially in the "wings" of the attractor. By contrast, LSTM-29 significantly outperforms both the imperfect and LSTM-0 models in almost all regions, with the exception of the attractor's boundaries. This discrepancy may stem from the relative scarcity of data at the boundaries -a common issue with the Lorenz 63 system- making it challenging for the LSTM to improve in these areas.

As the forecast horizon increases to 2–4*dt*, a similar pattern emerges. LSTM-0 continues to underperform in the wings of the attractor relative to the imperfect model but demonstrates strong performance at the bifurcation point, where it surpasses the imperfect model. By the final iteration, LSTM-29 exhibits substantial improvements over LSTM-0, reducing the MFE across




the attractor. For 2*dt* forecasts, the imperfect model still performs better along the attractor's boundaries, but for 3–4*dt* forecasts, LSTM-G29 and the imperfect model exhibit comparable performance in these regions.

**Discussion**

This paper demonstrates that cycling through training data to generate and refine DDMs offers a viable approach for
improving model accuracy. This approach is particularly relevant in scenarios where imperfect models represent complex systems. By leveraging DDMs, it becomes possible to capture missing system dynamics, resulting in a more accurate and robust system representation. The methodology presented here underscores the potential of combining imperfect models with DA and iterative refinements using DDMs.

Our results highlight the power of this approach. For shorter forecast windows, significant improvements were observed after four iterations of the algorithm, reducing both analysis and forecast errors compared to the baseline imperfect model. For longer forecast windows, the iterative cycling algorithm demonstrated similar gains, albeit at a slower rate of improvement. Across all iterations and forecast horizons, training the LSTM with cycling consistently enhanced both the analysis and forecast trajectories. This underscores the strength of the proposed method in capturing complex system
dynamics and refining forecasts over successive iterations.

The implications for real-world applications are significant. DDMs have become indispensable in a variety of fields, including weather forecasting, climate modelling, financial systems, and healthcare analytics. Our findings suggest that cycling over training using DA substantially improve the accuracy of DDMs in these contexts. This offline training
approach, while computationally intensive, has the advantage of allowing iterative refinement without requiring additional observations or real-time adjustments. The trade-off, however, is the increased computational cost associated with running multiple iterations of the algorithm.

Future work could focus on optimising the computational efficiency of this method to expand its applicability. For instance,
exploring hybrid approaches that combine traditional ensemble techniques or reduced-order models with cycling could alleviate some of the computational burden. Additionally, testing this framework on more diverse systems, such as high-dimensional chaotic systems, would provide deeper insights into its scalability and robustness. Finally, developing adaptive cycling algorithms that adjust the number of iterations dynamically based on real-time error metrics could strike a balance between performance gains and computational demands.

In summary, our research highlights the potential of iteratively improving DDMs through a cyclic approach. By combining DDMs, DA, and iterative refinement, we provide a pathway to enhancing model accuracy in complex systems—a critical step toward more reliable and robust predictions across a wide range of domains.



**Author contributions**

T.M. conceptualised research; M.G. and T.M. designed experiments; M.G. performed experiments; M.G. and T.M. analysed data; and M.G. and T.M. wrote the paper.

**Acknowledgments**

This work was supported by JST, SATREPS Grant Number: JPMJSA2109, JSPS KAKENHI Grant Number JP24H00021,
Japan Aerospace Exploration Agency, the COE research grant in computational science from Hyogo Prefecture and Kobe City through Foundation for Computational Science, and the RIKEN Pioneering Project "Prediction for Science".

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
