# Peer review of "Using Data Assimilation to Improve Data-Driven Models."

_EGUsphere, 2025_

## Referee Comment (RC1)

Review of *Using Data Assimilation to Improve Data-Driven Models*
by Michael Goodliff and Takemasa Miyoshi
Sumitted to Nonlin. Processes Geophys.
Egusphere-2025-933. Jie Feng Editor.

Marc Bocquet

April 3, 2025

**1   Main remarks**

This manuscript could have made a very nice paper. As explained by Zheqi Shen, acting as first reviewer, it would nonetheless require a more refined and careful writing and more thorough experiments. However, in my opinion, the main problem comes from the fact that the arguably nice idea which consists in using data assimilation (DA) and machine learning (ML) to iteratively improve a dynamical model, has already been put forward, extensively explored and even mathematically justified in the literature over the past six years, in a stream of papers. In that regard, the manuscript misses almost all of the literature on the topic. It turns out that this idea has been proposed by colleagues of mine and myself, so that I can provide a brief summary of what has been investigated on the topic. However, at Riken Center for Computational Science, you may have the computational resources to further test the concept on challenging high-dimensional geofluid models. Here is my own take on the subject's developments:

- To the best of my knowledge, the idea of combining DA and ML, from an accessible standpoint by the DA community, has been introduced in Hsieh and Tang (1998) and further discussed by Abarbanel et al. (2018) and then Bocquet et al. (2019). In the last two references, the general approach and its related key cost function is mathematically justified by a Bayesian derivation.

- The numerical solution of the combined DA and ML problem – a far from trivial heterogeneous optimisation problem – was explored in Bocquet et al. (2019). This approach was shown to be a viable way to assimilate partial and noisy observations in order to estimate ML dynamical models. In this specific paper the neural network parameters and the state control variables are solved for altogether without splitting the DA and ML steps iteratively. It was extensively tested on the Lorenz 1963, the Kuramoto-Sivashinsky, and the two-scale Lorenz models.

- In Brajard et al. (2020), that you misquoted with a preprint reference, the iterative scheme to solve the same loss function is first proposed and tested on the Lorenz 1996 model. In Bocquet et al. (2020), the whole theory is justified and shown to be equivalent to a coordinate descent minimisation of the Bayesian cost function mentioned above. It is extensively discussed and the expectation-maximisation statistical technique is further combined to the method to estimate the residual model error. Both are highly cited papers.

- The case where one starts the iterations with an imperfect physical model has then been extensively discussed in Brajard et al. (2021); Farchi et al. (2021b). This has been tested on the two-scale Lorenz model, the MAOOAM climate low-order model, and the ECMWF quasi-geostrophic model.

- This stream of ideas and papers has been summarised and further discussed in a perspective paper (Bocquet, 2023).

- These ideas were finally generalised from offline to potentially online training, and applied by Farchi et al. (2021a, 2023) to increasingly complex models, and it was shown how they could be integrated in the operational data assimilation IFS-based workflow at the ECMWF. Finally it was successfully tested in a pre-operational IFS configuration in Farchi et al. (2025). This corresponds to implementing the iterative loop at order 3/2, in the terminology introduced by Bocquet (2023).

In the field of sea-ice modelling, Gregory et al. (2024) also recently provided a nice example of the iterative scheme.

**2    Suggestions and typos:**

1. l.43-47: As mentioned above, the reference is just a preprint; the corresponding published paper is Brajard et al. (2020). How to improve an already existing imperfect model is investigated in, e.g., Brajard et al. (2021); Farchi et al. (2021b).

2. l.62: "Lorenz 63 Systems" $\longrightarrow$ "Lorenz 63 models"

3. l.63: "two version" $\longrightarrow$ "two versions"

4. l.65: "which come" $\longrightarrow$ "which comes"

5. l.79: $0 \dots 5 \longrightarrow 0 \dots 4$.

6. In order to help the readers of scientific papers, it is highly recommended to number the equations.

7. Equation at l.90: In classical data assimilation, and quite often in the geoscientific literature, vectors and matrices are bold, which is not the case here. This also applies to the subsequent equations, with further inconsistencies in the notation.

8. l.106 and l.110: "The Background/Observation Term" $\longrightarrow$ 'The background/observation term".

9. The introduction to 3D-Var might be a little too plain for a DA paper.

10. l.144: "of the system" $\longrightarrow$ "of the model".

11. l.158: "In contrast," $\longrightarrow$ "By contrast,".

12. l.239-244: Most of the leads that you proposed in this section of the conclusion have actually been explored (and more) in the papers that I mentioned above.

**References**

Abarbanel, H.D.I., Rozdeba, P.J., Shirman, S., 2018. Machine learning: Deepest learning as statistical data assimilation problems. Neural Computation 30, 2025–2055. doi:10.1162/neco_a_01094.

Bocquet, M., 2023. Surrogate modelling for the climate sciences dynamics with machine learning and data assimilation. Front. Appl. Math. Stat. 9. doi:10.3389/fams.2023.1133226.

Bocquet, M., Brajard, J., Carrassi, A., Bertino, L., 2019. Data assimilation as a learning tool to infer ordinary differential equation representations of dynamical models. Nonlin. Processes Geophys. 26, 143–162. doi:10.5194/npg-26-143-2019.

Bocquet, M., Brajard, J., Carrassi, A., Bertino, L., 2020. Bayesian inference of chaotic dynamics by merging data assimilation, machine learning and expectation-maximization. Found. Data Sci. 2, 55–80. doi:10.3934/fods.2020004.

Brajard, J., Carrassi, A., Bocquet, M., Bertino, L., 2020. Combining data assimilation and machine learning to emulate a dynamical model from sparse and noisy observations: a case study with the Lorenz 96 model. J. Comput. Sci. 44, 101171. doi:10.1016/j.jocs.2020.101171.

Brajard, J., Carrassi, A., Bocquet, M., Bertino, L., 2021. Combining data assimilation and machine learning to infer unresolved scale parametrisation. Phil. Trans. R. Soc. A 379, 20200086. doi:10.1098/rsta.2020.0086.

Farchi, A., Bocquet, M., Laloyaux, P., Bonavita, M., Malartic, Q., 2021a. A comparison of combined data assimilation and machine learning methods for offline and online model error correction. J. Comput. Sci. 55, 101468. doi:10.1016/j.jocs.2021.101468.

Farchi, A., Chrust, M., Bocquet, M., Bonavita, M., 2025. Development of an offline and online hybrid model for the integrated forecasting system. Q. J. R. Meteorol. Soc. , e4934URL: `https://rmets.onlinelibrary.wiley.com/doi/abs/10.1002/qj.4934`, doi:10.1002/qj.4934.

Farchi, A., Chrust, M., Bocquet, M., Laloyaux, P., Bonavita, M., 2023. Online model error correction with neural networks in the incremental 4D-Var framework. J. Adv. Model. Earth Syst. 15, e2022MS003474. doi:10.1029/2022MS003474.

Farchi, A., Laloyaux, P., Bonavita, M., Bocquet, M., 2021b. Using machine learning to correct model error in data assimilation and forecast applications. Q. J. R. Meteorol. Soc. 147, 3067–3084. doi:10.1002/qj.4116.

Gregory, W., Bushuk, M., Zhang, Y., Adcroft, A., Zanna, L., 2024. Machine learning for online sea ice bias correction within global ice-ocean simulations. Geophys. Res. Lett. 51, e2023GL106776. doi:10.1029/2023GL106776.

Hsieh, W.W., Tang, B., 1998. Applying neural network models to prediction and data analysis in meteorology and oceanography. Bull. Amer. Meteor. Soc. 79, 1855–1870. doi:10.1175/1520-0477(1998)079¡1855:ANNMTP¿2.0.CO;2.